# Income Disparities in Obesity Trends among U.S. Adults: An Analysis of the 2011–2014 California Health Interview Survey

**DOI:** 10.3390/ijerph19127188

**Published:** 2022-06-11

**Authors:** Shaoqing Gong, Liang Wang, Zhongliang Zhou, Kesheng Wang, Arsham Alamian

**Affiliations:** 1School of Public Policy and Administration, Xi’an Jiaotong University, Xi’an 710049, China; zzliang1981@xjtu.edu.cn; 2Department of Public Health, Robbins College of Health and Human Science, Baylor University, Waco, TX 76789, USA; Liang_Wang1@baylor.edu; 3Department of Family and Community Health, School of Nursing, West Virginia University, Morgantown, WV 26506, USA; kesheng.wang@hsc.wvu.edu; 4School of Nursing and Health Studies, University of Miami, Coral Gables, FL 33146, USA; arsham.alamian@miami.edu

**Keywords:** obesity, income disparities, epidemiology

## Abstract

The aim of this study was to examine income disparities in obesity trends among California adults. Data were obtained from the 2011–2014 California Health Interview Survey (*n* = 83,175 adults). Obesity for adults was defined as a body mass index of 30 kg/m^2^ or above. Family income was categorized as below 100%, 100% to 299%, or 300% and above of the federal poverty level (FPL). Weighted multiple logistic regression analyses were used to examine the association between family income and obesity across survey years after controlling for age, sex, race/ethnicity, smoking status, marital status, education, physical activity, and healthy diet. Obesity prevalence among California adults increased slightly from 25.1% in 2011 to 27.0% in 2014. Compared to 300% FPL or above, <100% FPL and 100–299% FPL were associated with increased odds of obesity, respectively (OR = 1.35, 95% CI = 1.22–1.50, for 100–299% FPL; OR = 1.18, 95% CI = 1.10–1.27, for 300% FPL or above). Each year, lower FPL was associated with higher odds of obesity, except for the year 2014. An inverse association between obesity and family income in each survey year was observed, with the magnitude of the income disparity decreasing from 2011 to 2014. The findings of this study show that family income was negatively associated with obesity among adults in California from 2011–2014, and the magnitude of the income disparity in obesity prevalence decreased over this period. Future studies need to examine potential risk factors associated with the decreasing trend.

## 1. Introduction

Obesity has been a public health crisis in the United States (U.S.), because the age-adjusted prevalence of obesity in U.S. adults was 42.4% in 2017–2018 [1]. Obesity disparities are complicated and vary by sex, age, race/ethnicity, and socioeconomic status (SES) [2]. Some subgroups of the population have a higher prevalence of obesity, such as those at low SES or minority groups [3]. Several national surveys, including the National Health and Nutrition Examination Survey (NHANES) and the Early Childhood Longitudinal Study (ECLS), demonstrate obvious disparities by ethnicity. According to NHANES 2017–2018 data, the prevalence of obesity was lowest among non-Hispanic Asian adults (17.4%) compared with non-Hispanic white (42.2%), non-Hispanic black (49.6%), and Hispanic (44.8%) adults. Non-Hispanic black adults had the highest prevalence of obesity compared with all other race and Hispanic-origin groups [1]. In addition, ECLS data show that American Indians had the highest prevalence (37%), and Asians had the lowest prevalence (15.8%) among preschool U.S. boys [4].

Compared to a number of studies examining the ethnic differences in obesity, fewer studies have reported SES disparity in obesity prevalence. The 2001–2007 California Health Interview Survey (CHIS) examining obesity disparities by family income and gender found that obesity prevalence increased between 2001 and 2007 among lower-income adolescents, but not among higher-income adolescents. Overall, family income disparity in obesity doubled in this period, but trends were more consistent among males than among females [5]. Between 2003 and 2008, a decrease in obesity prevalence among children aged <6 years was observed in eastern Massachusetts [6]. The Pediatric Nutrition Surveillance System during 2008–2011 included about 11.6 million low-income children aged 2–4 years with measured weight and height. It showed that obesity rates among low-income preschoolers modestly declined in 19 U.S. states and territories [7], but this decline may vary by SES.

The U.S. Department of Agriculture shows that 9.7% of the U.S. population (i.e., 29.7 million people) live in low-income areas. These areas refer to locations that are more than one mile from a supermarket, where the only choice to buy foods includes “convenience” stores, liquor stores, gas stations, or fast-food restaurants [8]. Families with low incomes are less likely to own a car, and thus may be more likely to consume shelf-stable food [9], which contain refined grains, added sugars, and added fats. Income level plays a role in an individual’s food consumption and physical activity patterns. For example, fresh vegetables and fruits usually cost more than fast food, and sometimes, healthy foods or healthy super markets are not available in poor neighborhoods, thus influencing food access. Neighborhoods with low income also negatively affect opportunities for physical activity because of fewer playgrounds, sidewalks, and recreational facilities [10].

Obesity disparities in U.S. adults suggest that obesogenic environmental changes may likely influence some groups more than others. This is because different groups may have different responses to the environmental factors. Reducing the obesity disparities in their risk factors can be targeted [11]. More research and programs are needed to help better understand the causes of obesity disparities. Despite an earlier report for income disparities in obesity trends in California adolescents [5], the present study aims to use 2011–2014 CHIS to examine if income disparities exist in obesity trends among California adults, and if the observed disparities differ by socio-demographic factors.

## 2. Methods

### 2.1. Study Population

The CHIS is the U.S. largest state health survey and a critical source of data on Californians, as well as on the state’s various racial and ethnic groups. A two-stage, geographically stratified design was used to produce a representative sample of the state. Residential telephone numbers were selected from within predefined geographic areas, and respondents were then randomly selected from within sampled households. For nearly two decades, CHIS has been collecting information via a random-digit-dial telephone survey on a noninstitutionalized population-representative sample of California’s adults, adolescents, and children. This study restricted analysis for the adults only. These telephone interviews allow the CHIS to track important health conditions and health behaviors in California. To ensure coverage of respondents with limited English proficiency, in addition to English, CHIS is administered in Spanish, Cantonese, Mandarin, Korean, Tagalog, and Vietnamese, to allow a large sample of a diverse population; therefore, CHIS data have great ability to report on racial/ethnic differences.

The large CHIS sample includes people from many ethnic groups to provide health-related information for most large and small racial and ethnic populations that are all a part of California. CHIS telephone surveys are conducted in all 58 counties of California. The CHIS may conduct oversampling of specific urban areas, such as Los Angeles and San Diego. In this study, the sample sizes for year 2011, 2012, 2013, and 2014 were 22,580, 20,355, 20,724, and 19,516, respectively, which were used for analyses due to non-missing data.

### 2.2. Study Variables

#### 2.2.1. Family Income

Family income was reported by the adult respondent, usually a parent, and was examined as a percentage of the federal poverty level, which adjusts for total household income and number of members in the household. Family income was categorized as below 100%, 100% to 299%, or 300% and above the federal poverty level (FPL).

#### 2.2.2. Obesity

Body mass index (BMI: weight in kilograms divided by height in meters squared) calculations were based on adults’ self-reported height and weight, and divided into four categories: obesity, overweight, healthy weight, and underweight. Obesity for adults was defined as a BMI of 30 kg/m^2^ or above. In this study, obesity was dichotomized into obese and nonobese.

#### 2.2.3. Covariates

Demographic variables included self-reported age, sex, and race/ethnicity (Latino, White, Asian, African American, American Indian, or multiple races). Smoking status was defined as current smoker or not current smoker. Marital status was defined as married, never married, or other. Education attainment was included in three categories as high school, college, or graduate. Physical activity was defined as walking at least 10 min for either transportation or leisure over the past 7 days. Healthy diet was determined by the response to the following question: “how often find fresh fruit/vegetable in neighborhood?” Response categories included: never, sometimes or usually, and always.

### 2.3. Statistical Analysis

Characteristics of California adults were described according to demographics, lifestyle risk factors, family income, and weight status. Obesity prevalence among adults was examined by survey year and family income from 2011–2014. Multiple logistic regression analyses were used to examine the association between family income and obesity across survey years after controlling for age, sex, race/ethnicity, smoking status, physical activity, healthy diet, marital status, and education attainment. All analyses were weighted to be representative of the California population, and were adjusted for the complex survey design of the CHIS. All analyses were two-sided and performed with SAS version 9.1 (SAS Institute Inc., Cary, NC, USA).

### 2.4. Statistical Power Analysis

We used G*Power v.3.1.9.4 to compute the power for the present study [12,13]. Based on the sample size of 83,175, given type 1 error rate (α = 0.05), small effect size (odds ratio = 1.5), using z-test in logistic regression model with multiple predictors, R^2^ (the percentage of variance explained by the predictors) of other predictors = 0.1, the power could reach 99.99%.

## 3. Results

Table 1 describes the participants’ characteristics of adults in California from 2011 to 2014. In each year, about half of participants were at 300% FPL or above. With regard to race/ethnic distribution, the majority were whites, followed by Latinos, Asians, and African Americans. The prevalence of overweight and obesity was high, accounting for over one third and about one fourth of total population, respectively, at each time period. The majority of people were not current smokers, were physically active, and reported finding always fresh fruit/vegetable in their neighborhood. Approximately half of California adults were married and had a college degree.

Table 2 shows adjusted odds ratios for the association between family income and obesity in Californian adults using multiple logistic regression analyses. Overall, compared to being at 300% FPL or above, being at <100% FPL or at 100–299% FPL were associated with increased odds of obesity (OR = 1.35, 95% CI = 1.22–1.50, for <100% FPL; OR = 1.18, 95% CI = 1.10–1.27, for 100–299% FPL or above). Each year, lower FPL was associated with higher odds of obesity, except the year 2014. In addition, Asians had lower odds of obesity than whites in the overall sample (OR = 0.41, 95% CI = 0.35–0.48), whereas Latinos, African Americans, and American Indians had higher odds of obesity than whites (OR = 1.62, 95% CI = 1.49–1.75; OR = 2.12, 95% CI = 1.87–2.41; OR = 2.04, 95% CI = 1.52–2.74, respectively). Furthermore, education and physical activity were both found to be significantly associated with obesity.

Figure 1 examines the prevalence of obesity in adults by survey year and family income from 2011–2014. Overall, obesity prevalence among California adults increased slightly from 25.1% in 2011 to 27.0% in 2014. An inverse association between obesity and family income in each survey year was observed in bivariate analyses, with the magnitude of the income disparity decreasing from 2011 to 2014. Specifically, in 2011, obesity prevalence was 50.2% higher among adults with family incomes below the poverty line (0–99% FPL) than among those whose family incomes were 300% or more of the FPL (32% vs. 21.3%; *p* < 0.0001). In 2014, obesity prevalence was 33.6% higher in the lowest income group than in the highest income group (31.4% vs. 23.5%; *p* < 0.0001).

## 4. Discussion

With obesity’s sustained prevalence in the U.S. and around the globe in recent years, it is imperative to elucidate the driving factors that contribute to this disease. Understanding the disease distribution of burden among varying communities and the present social inequities can provide direction for future efforts. Regarding income disparity, it is a two-pronged issue in relation to the economic development of a country, as well as an individual’s SES within each of those countries.

The economic development of a country has been shown in recent literature to provide some insight into obesity trends. For instance, those who live in low-income countries and fall into high-SES criteria have a higher probability of being obese [14]. However, in contrast, those who live in high-income countries and fall into high-SES criteria, have a higher likelihood of not being obese [14]. These trends may be based on a multitude of factors. Some may include cultural differences, lifestyle, and dietary habits/available foods within the region. Though the income status of the country a person is living in seems to be something to take note of, an individual’s personal SES has also been found to be a determining factor.

This study used data of California adults to show that lower family income was positively associated with obesity from 2011 to 2014. The findings for the association among overall population were similar when each survey cycle was examined. However, the magnitude of the income disparity decreased during this period, which is different from a study using CHIS from 2001–2007 [15]. In that study, differential trends were observed in obesity over time among lower-income children relative to higher-income adults. It was hypothesized that obesity prevalence likely increased more among lower-income adults than among higher-income adults, which is inconsistent with our findings, i.e., obesity prevalence has increased more among those with higher incomes. Although people with high incomes are likely to have access to greater resources that may better protect them from preventable conditions, they may not behave as expected to consume healthy foods. Despite a reduced magnitude of the income disparity, the prevalence of obesity remains higher in adults with lower family income than those with higher family income in each survey year; and adjusted analyses confirm an inverse association between family income and obesity, suggesting a necessity of ameliorating poverty status to have better access to healthy foods.

Several previous studies have examined the association between family income and obesity among youth [16,17] without considering SES differences [18], and/or used old data [16,17]. The present study is among the first to examine income disparities in obesity trends among U.S. adults using more recent data with diverse races/ethnicities. The findings provide evidence to help future obesity intervention studies target family income. It would be interesting to explore the reasons of observed changes in obesity prevalence in this study, which may be due to changes in some risk factors of obesity. For example, physical activity level and consumption of healthy food may decrease, but smoking behavior may increase among adults with high incomes as compared to those with low incomes. In this study, the greatest increases in obesity were found among adults whose family incomes were above the poverty line, and there was a decrease among those below the poverty line. However, we only used the recent four years’ data, and need to examine SES-specific trends in obesity over a longer period of time.

A study was conducted in the U.S. to determine whether income inequality was associated with obesity status. The findings of this study proved to be enlightening, as they determined varying facets for targeted interventions. Different factors they found to be driving the strong association were sex, economically diverse/homogenous location, and income itself. The first of which was that statistically significant associations were found more often among men with middle-to-high income in comparison to lower-income men [19]. This association was not replicated among women. However, women who were obese had a higher income inequality when compared to their counterparts (men) [19]. Another factor to note was the independent effect of living in economically diverse communities versus those who lived in financially homogeneous locations [19]. Based on these findings, in the U.S., women of low SES and men of all income levels should be prioritized for future programs and interventions to reduce obesity. Specifically, work should be done to create accessible, healthy foods among those in vulnerable populations and low-income communities, as well as work to create sustainable health practices.

Using the same data set, Zare et al. investigated whether the relationship between obesity and income differed by sex and race. They found that the poverty income ratio and obesity were significant and positive more often among white non-Hispanics and black non-Hispanics with middle and high income than those in the lower income category; this association was not found among Mexican Americans [20]. When considering obese women, compared to white women, Mexican American and black women faced higher income inequality [20]. In looking at men, black non-Hispanic males were found to have the highest income inequality [20]. So, besides looking at addressing the systematic failures of income distribution, it is also imperative to take note of sex, race/ethnicity, and economically homogeneous/heterogeneous locations. This continues to be a complex public health problem, so understanding what specific factors contribute to income disparities and, ultimately, obesity remain paramount.

This study has some limitations. First, the cross-sectional survey data used in this study cannot establish causality. Second, height and weight were self-reported rather than measured, which may have likely introduced a measurement bias. Although it has been suggested that self-reported BMI has been widely used in population-based surveys [15,21,22,23], it may result in underestimated BMIs [21,22]. Third, the statistical analysis does not demonstrate the variables normality, and neither does the sample calculation. Fourth, CHIS is a telephone survey, and the interview completion rates declined from 2011 to 2014, which is a little old to represent the current trend. Nevertheless, it is representative of the California population, and the response rate has not been demonstrated to introduce a nonresponse bias [24]. Fifth, Figure 1 examined the association between income and obesity by survey year, which could be further adjusted for some more covariates. However, fully adjusted results are shown in Table 2 from multiple logistic regression analysis. Sixth, health diet and physical activity were only presented by one question. Health diet may also consider the food resource or fast-food consumption, which may compromise the validity of the health diet question. Seventh, the findings regarding the obesity trend need to be interpreted with caution. Specifically, this study applied multiple logistic regression models to estimate the OR for obesity in the lower-income groups compared with the higher-income group annually (i.e., 2011, 2012, 2013, and 2014). However, without considering time-related factors, the question of whether income influences obesity trends may not be appropriately answered. Furthermore, samples might vary year by year, and this heterogeneity should be considered in the model in future studies. Eighth, this study is based on observational data, and there may be unobserved confounders that are not included in the analyses, leading to residual confounding issues. Finally, the results from California cannot be used to represent the nation. However, California has the population with the most diverse races/ethnicities, and spends more public and private money on the health consequences of obesity than other states [25]. Therefore, findings can also help adults in other states pay attention to this public health issue.

## 5. Conclusions

The findings of this study show that lower family income was positively associated with obesity among adults in California from 2011 to 2014, and the magnitude of the income disparity in obesity prevalence decreased over this period. Future studies need to examine potential reasons for the decreasing trend. Studies using national data and longitudinal designs are needed to confirm causality.

## Figures and Tables

**Figure 1 ijerph-19-07188-f001:**
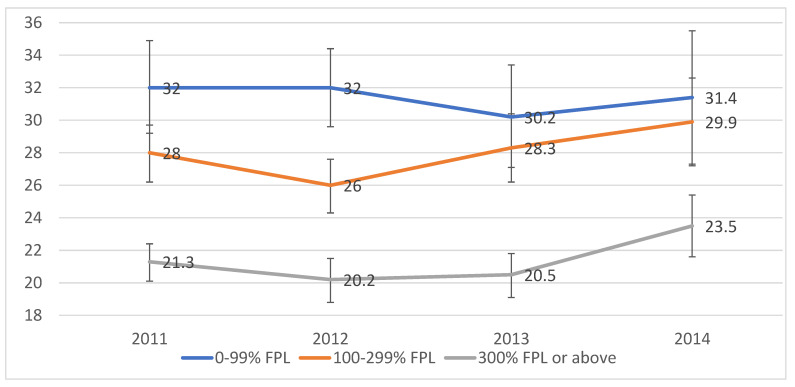
Obesity prevalence among adults by survey year and family income: California Health Interview Survey, 2011–2014. Note. FPL = federal poverty level. Survey weights were used in all analyses, and analyses were adjusted for the complex survey design. Obesity was defined as a body mass index of 30 or above.

**Table 1 ijerph-19-07188-t001:** The participants’ characteristics of Californian adults by each survey year from 2011–2014.

	2011 (*n* = 22,580)	2012 (*n* = 20,355)	2013 (*n* = 20,724)	2014 (*n* = 19,516)
**Family income, *n* (%)**				
<100% FPL	2898 (15.3)	3374 (17.3)	2555 (15.2)	2753 (16.4)
100–299% FPL	7197 (33.2)	6825 (34.0)	6743 (34.7)	6410 (35.1)
≥300% FPL	12,485 (51.5)	10,156 (48.7)	11,426 (50.1)	10,353 (48.4)
**Age, mean (SD)**	43.2 (0.02)	43.2 (0.03)	43.4 (0.03)	43.6 (0.03)
Sex, *n* (%)				
Female	13,089 (51.1)	11,998 (51.3)	12,195 (51.3)	11,627 (51.2)
Male	9491 (48.9)	8357 (48.7)	8529 (48.7)	7889 (48.8)
**Race/ethnicity, *n* (%)**				
Latino	4555 (33.3)	4951 (34.2)	4203 (34.7)	3793 (34.9)
White	14,471 (45.5)	11,272 (43.4)	13,324 (42.9)	12,319 (42.3)
Asian	1752 (13.1)	2474 (13.8)	1491 (13.8)	2003 (14.0)
African American	1095 (5.6)	902 (5.6)	979 (5.6)	785 (5.6)
American Indian	161 (0.7)	304 (0.4)	165 (0.4)	145 (0.5)
Mixed race or other	546 (2.0)	452 (2.5)	562 (2.6)	471 (2.7)
**Weight status, *n* (%)**				
Underweight	389 (1.6)	358 (1.6)	324 (1.3)	345 (1.6)
Healthy weight	8509 (38.1)	7801 (39.1)	7735 (38.0)	7168 (35.8)
Overweight	7931 (35.2)	7077 (35.1)	7398 (36.0)	6848 (35.5)
Obesity	5751 (25.1)	5119 (24.2)	5267 (24.7)	5155 (27.0)
**Smoking status, *n* (%)**				
Current smoker	2717 (14.3)	2260 (13.2)	2279 (13.0)	1843 (11.7)
Not current smoker	19,863 (85.7)	18,095 (86.8)	18,445 (87.0)	17,673 (88.3)
**Marital status, *n* (%)**				
Married	10,909 (51.8)	10,452 (50.6)	10,105 (50.4)	9782 (50.3)
Never married	7545 (23.1)	6563 (24.0)	7116 (22.9)	6876 (23.1)
Others	4126 (25.1)	3340 (25.4)	3503 (26.6)	2858 (26.6)
**Education, *n* (%)**				
High school	7391 (40.2)	7453 (40.3)	6482 (39.8)	6342 (39.4)
College	11,610 (47.0)	9826 (47.3)	10,538 (46.9)	9745 (47.6)
Graduate	3579 (12.8)	3076 (12.4)	3704 (13.3)	3429 (13.0)
**Physical activity ^a^ *n* (%)**				
Yes	17,258 (80.4)	15,641 (79.3)	15,671 (79.8)	14,582 (80.2)
No	5322 (19.6)	4714 (20.7)	5053 (20.2)	4934 (19.8)
**Heathy diet ^b^ *n* (%)**				
Never	1237 (4.5)	1256 (4.8)	1094 (4.8)	958 (4.1)
Sometimes/usually	3271 (17.4)	2999 (17.2)	3085 (19.6)	3188 (19.8)
Always	18,072 (78.2)	16,100 (77.9)	16,545 (75.7)	15,370 (76.1)

Abbreviation: FPL = federal poverty level; SD, standard deviation. Note: estimates were weighted to be representative of the California population, and were adjusted for complex survey design effects. ^a^ Physical activity was defined as walking at least 10 min for either transportation or leisure over the past 7 days. ^b^ Healthy diet was determined by response to “how often find fresh fruit/vegetable in neighborhood?”, including three categories: never, sometimes or usually, and always.

**Table 2 ijerph-19-07188-t002:** Logistic regression analysis for the association between family income and obesity (vs. absence of obesity) among adults in California, 2011–2014.

Characteristics	Overall	2011	2012	2013	2014
	AOR (95% CI)	AOR (95% CI)	AOR (95% CI)	AOR (95% CI)	AOR (95% CI)
**Family income**					
<100% FPL	1.35 (1.22–1.50) ***	1.40 (1.17–1.68) ***	1.46 (1.22–1.74) ***	1.37 (1.13–1.66) **	1.19 (0.92–1.55)
100–299% FPL	1.18 (1.10–1.27) ***	1.19 (1.07–1.34) **	1.11 (0.97–1.27)	1.31 (1.14–1.51) ***	1.13 (0.95–1.34)
≥300% FPL	Referent	Referent	Referent	Referent	Referent
**Age in years**	1.01 (1.00–1.01) ***	1.01 (1.01–1.01) ***	1.01 (1.00–1.01) ***	1.01 (1.00–1.01) **	1.01 (1.00–1.01) ***
Sex					
Male	1.15 (1.09–1.22) ***	1.19 (1.08–1.31) ***	1.10 (0.97–1.23)	1.18 (1.04–1.32) **	1.15 (1.01–1.31) *
Female	Referent	Referent	Referent	Referent	Referent
**Race/ethnicity**					
Latino	1.62 (1.49–1.75) ***	1.53 (1.36–1.73) ***	1.69 (1.45–1.98) ***	1.70 (1.44–2.00) ***	1.55 (1.33–1.82) ***
White	Referent	Referent	Referent	Referent	Referent
Asian	0.41 (0.35–0.48) ***	0.39 (0.32–0.49) ***	0.40 (0.29–0.55) ***	0.38 (0.27–0.55) ***	0.47 (0.33–0.66) ***
African American	2.12 (1.87–2.41) ***	1.91 (1.61–2.27) ***	2.17 (1.74–2.70) ***	2.24 (1.72–2.90) ***	2.20 (1.60–3.04) ***
American Indian	2.04 (1.52–2.74) ***	1.49 (0.90–2.46)	2.79 (1.78–4.36) ***	1.53 (0.69–3.38)	2.84 (1.38–5.88) **
Mixed race or other	1.14 (0.94–1.39)	1.55 (1.15–2.10) **	1.02 (0.66–1.57)	1.26 (0.85–1.85)	1.84 (0.76–4.45)
**Smoking status**					
Current smoker	0.94 (0.85–1.03)	0.94 (0.81–1.10)	0.92 (0.74–1.13)	1.05 (0.88–1.25)	0.87 (0.67–1.11)
Not current smoker	Referent	Referent	Referent	Referent	Referent
**Marital status**					
Married	Referent	Referent	Referent	Referent	Referent
Never married	0.75 (0.69–0.83)	0.82 (0.71–0.94) **	0.67 (0.56–0.81) ***	0.77 (0.62–0.94) *	0.78 (0.62–0.98) *
Others	1.01 (0.93–1.09)	1.00 (0.90–1.10)	0.95 (0.82–1.11)	0.92 (0.79–1.08)	1.18 (0.99–1.41)
**Education**					
High school	1.48 (1.33–1.65) ***	1.63 (1.39–1.92) ***	1.62 (1.30–2.03) ***	1.19 (0.96–1.47)	1.54 (1.22–1.95) ***
College	1.34 (1.21–1.49) ***	1.43 (1.23–1.67) ***	1.50 (1.22–1.85) ***	1.08 (0.89–1.32)	1.40 (1.12–1.75) **
Graduate	Referent	Referent	Referent	Referent	Referent
**Physical activity ^a^**					
Yes	0.70 (0.65–0.75) ***	0.78 (0.70–0.87) ***	0.69 (0.60–0.80) ***	0.63 (0.55–0.74) ***	0.70 (0.60–0.81) ***
No	Referent	Referent	Referent	Referent	Referent
**Healthy diet ^b^**					
Never	1.05 (0.92–1.19)	0.94 (0.74–1.19)	1.17 (0.93–1.48)	1.04 (0.77–1.39)	1.06 (0.77–1.47)
Sometimes/usually	0.93 (0.85–1.03)	0.92 (0.80–1.07)	0.99 (0.82–1.20)	0.89 (0.72–1.11)	0.95 (0.77–1.17)
Always	Referent	Referent	Referent	Referent	Referent

Abbreviations: AOR, adjusted odd ratio; CI, confidence interval; PFL, poverty federal level. ^a^ Physical activity was defined as walking at least 10 min for either transportation or leisure over the past 7 days. ^b^ Healthy diet was determined by response to “how often find fresh fruit/vegetable in neighborhood?”, including three categories: never, sometimes or usually, and always. * <0.05, ** <0.01, *** <0.001.

## Data Availability

Data available through the CHIS Online Portal: https://healthpolicy.ucla.edu/chis/Pages/default.aspx (accessed on 1 October 2021).

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
