# Peer review of "Income Disparities in Obesity Trends among U.S. Adults: An Analysis of the 2011–2014 California Health Interview Survey"

_ijerph, 2022, doi:10.3390/ijerph19127188_

Round 1

Reviewer 1 Report

A relevant study, that examined income disparities in obesity trend among California adults. This study observed disparities differ by socio-demographic factors in obesity prevalence.

This study has some important points, emphasizing the inclusion of diverse race/ethnic groups in 58 counties of California.

But this study has some limitations, first the temporal issue since the data are nor current. The statistical analysis does not demonstrate the variables normality, neither the sample calculation.

Author Response

  1. A relevant study, that examined income disparities in obesity trend among California adults. This study observed disparities differ by socio-demographic factors in obesity prevalence. This study has some important points, emphasizing the inclusion of diverse race/ethnic groups in 58 counties of California.

Response:

Thank you.

  1. But this study has some limitations, first the temporal issue since the data are nor current. The statistical analysis does not demonstrate the variables normality, neither the sample calculation.

Response:

Thanks for the suggestions. We performed power calculation using G*Power software and added power analysis in the method. We also added some limitations in the discussion section.

References:

Faul F, Erdfelder E, Lang AG, Buchner A. G*Power 3: A flexible statistical power analysis program for the social, behavioral, and biomedical sciences. Behavior Research Methods, 2007; 39: 175-91.   

Faul F, Erdfelder E, Buchner A, Lang AG. (2009). Statistical power analyses using G*Power 3.1: Tests for correlation and regression analyses. Behavior Research Methods, 2009; 41: 1149-1160.

Reviewer 2 Report

  1. The survey data is a little old to be published.
  2. Fig1 showed the pattern of obesity prevalence among family income during 2011-2014. However, there were no covariates adjusted by Figure1. It is suggested that the author might perform a multiple logistic regression analysis to examine the association between obesity and survey year by three family income.
  3. Is it suitable that the “Health diet “ and “physical activity” only present by one question? “Health diet “ may also consider the food resource or fast food consumption, and I wonder the validity of “health diet “ question.
  4. The obesity in income disparities is an important and complex public health problem. I expect the author can provide more value discussion content in this paper.

Author Response

  1. The survey data is a little old to be published.

Response:

Thanks for the suggestions. We added some limitations in the discussion section.

  1. Fig1 showed the pattern of obesity prevalence among family income during 2011-2014. However, there were no covariates adjusted by Figure1. It is suggested that the author might perform a multiple logistic regression analysis to examine the association between obesity and survey year by three family income.

Response:

Thanks for the suggestions. Figure 1 aimed to examine the association between income and obesity by survey year, in which obesity is the outcome variable and income is the exposure variable. We did not perform the analysis to examine the association between obesity and survey year as the survey year then becomes the exposure variable, which is not our study aim. We added some limitations in the discussion section.

  1. Is it suitable that the “Health diet “and “physical activity” only present by one question? “Health diet “may also consider the food resource or fast food consumption, and I wonder the validity of “health diet” question.

Response:

Thanks for the suggestions. We added the limitations in the discussion section.

  1. The obesity in income disparities is an important and complex public health problem. I expect the author can provide more value discussion content in this paper.

Response:

Thanks for the suggestions. We added more discussions.

With obesity’s sustained prevalence in the U.S. and around the globe in recent years, it is imperative to elucidate the driving factors that contribute to this disease. Understanding the disease distribution of burden among varying communities and the present social inequities can provide direction for future efforts. Regarding income disparity, it is a two-pronged issue in relation to the economic development of a country as well as an individual’s SES within each of those countries.

The economic development of a country has been shown in recent literature to provide some insight into obesity trends. For instance, those who live in low-income countries and fall into high-socioeconomic criteria, have a higher probability of being obese (Houle, 2013). However, in contrast, those who live in high-income countries and fall into high-socioeconomic criteria, have a higher likelihood of not being obese (Houle, 2013). These trends may be based on a multitude of factors. Some may include cultural differences, lifestyle, and dietary habits/available foods within the region. While the income status of the country a person is living in seems to be something to take note of, an individual’s personal socioeconomic status has been found to be a determining factor.

A study was conducted in the U.S. to determine whether income inequality was associated with obesity status. The findings of this study proved to be enlightening as they determined varying facets for targeted interventions. Different factors they found to be driving the strong association were sex, economically diverse/homogenous location, and income itself. The first of which was that statistically significant associations were found more often among men in the middle- to high-income than in comparison to lower-income men (Zare et al., 2021a). This association was not replicated among women. However, women who were obese had a higher income inequality when compared to their counterparts (men) (Zare et al., 2021a). Another factor to note was the independent effect of living in economically diverse communities versus those who lived in financially homogeneous locations (Zare et al., 2021a). Based on these findings, in the US, women of low SES and men of all income levels should be prioritized for future programs and interventions to reduce obesity. Specifically, work should be done to create accessible, healthy foods among those in vulnerable populations and low-income communities and work to create sustainable health practices. 

Using the same data set, Zare et al. investigated whether the relationship between obesity and income differed by sex and race. They found that the poverty income ratio and obesity were significant and positive more often among white non-Hispanics and black non-Hispanics with middle and high income than those in lower income category; this association was not found among Mexican Americans (Zare et al., 2021b). When considering obese women, compared to white women, Mexican American and black women faced higher income inequality (Zare et al., 2021b). In looking at men, black non-Hispanic males were found to have the highest income inequality (Zare et al., 2021b). So not only looking at addressing systematic failures of the income distribution, but it is also imperative to take note of sex, race/ethnicity, and economically homogeneous/heterogeneous locations. This continues to be a complex public health problem so understanding what specific factors contribute to income disparities and, ultimately, obesity remains paramount.

References:

Houle B. (2013, December 3). How obesity relates to socioeconomic status. PRB. Retrieved May 9, 2022, from https://www.prb.org/resources/how-obesity-relates-to-socioeconomic-status/

Zare H, Gilmore DR, Creighton C, Azadi M, Gaskin DJ, Thorpe RJ. How income inequality and Race/Ethnicity Drive Obesity in U.S. adults: 1999–2016. Healthcare, 2021a; 9(11), 1442.

Zare H, Gaskin DD, Thorpe RJ. Income inequality and obesity among US adults 1999–2016: Does sex matter? International Journal of Environmental Research and Public Health, 2021b; 18(13): 7079.

Reviewer 3 Report

This study reports the obesity trends in California adults from 2011 to 2014, using samples of 19,516-22,580 per year and odds ratios across socioeconomic groups using multiple logistic regression analyses.

The first major issue is the conclusion about the obesity trend. Specifically, this study applied multiple logistic regression models to estimate the odds ratios for obesity in the lower-income groups compared with the higher-income group annually (i.e., 2011, 2012, 2013, and 2014).  However, without considering time-related factors, the question of whether income influences obesity trends cannot be properly answered. In Figure 1, there were no apparent within-group differences in obesity prevalence in 2011, 2012, and 2013. Even though the 0%-99% FPL group had a higher obesity prevalence in 2014, the error bars for obesity prevalence in 2011 and 2013 overlapped with the error bars in 2014. Furthermore, samples might vary year by year, and this heterogeneity should be considered in the model.

The second major problem is the rationale of why studying this population. It is unclear why the text in the Introduction section addresses obesity in children, not in adults. In addition, the references are out of date. The most recent one was published in 2014. The Introduction section should provide up-to-date background information clearly and a logical rationale for this study.

The third major problem is that the text used in the 2.1 section is highly similar to the text in their previous publication (https://pubmed.ncbi.nlm.nih.gov/30131498/). Although the study population might be the same, presenting in such a way may be regarded as self-plagiarism.

Author Response

  1. This study reports the obesity trends in California adults from 2011 to 2014, using samples of 19,516-22,580 per year and odds ratios across socioeconomic groups using multiple logistic regression analyses. The first major issue is the conclusion about the obesity trend. Specifically, this study applied multiple logistic regression models to estimate the odds ratios for obesity in the lower-income groups compared with the higher-income group annually (i.e., 2011, 2012, 2013, and 2014). However, without considering time-related factors, the question of whether income influences obesity trends cannot be properly answered. In Figure 1, there were no apparent within-group differences in obesity prevalence in 2011, 2012, and 2013. Even though the 0%-99% FPL group had a higher obesity prevalence in 2014, the error bars for obesity prevalence in 2011 and 2013 overlapped with the error bars in 2014. Furthermore, samples might vary year by year, and this heterogeneity should be considered in the model.

Response:

Thanks for the suggestions. We added the limitations in the discussion section.

  1. The second major problem is the rationale of why studying this population. It is unclear why the text in the Introduction section addresses obesity in children, not in adults. In addition, the references are out of date. The most recent one was published in 2014. The Introduction section should provide up-to-date background information clearly and a logical rationale for this study.

Response:

Thanks for the suggestions. We added the updated background information focusing on adult population in the introduction section.

  1. The third major problem is that the text used in the 2.1 section is highly similar to the text in their previous publication (https://pubmed.ncbi.nlm.nih.gov/30131498/). Although the study population might be the same, presenting in such way may be regarded as self-plagiarism.

Response:

Thanks for the suggestions. We revised accordingly.

Reviewer 4 Report

Thank you for letting me review this manuscript. The manuscript is well written in easily understandable. The presented data is rather old and statistical analysis is unspectacular but acceptable.

I think it must be more clear what the study finds.The Discussion says "This study used more recent data of California adults to show that lower family in-
come was positively associated with obesity from 2011 to 2014."

In the conclusion, the authors write "The findings of this study show that family income was negatively associated with obesity among adults in California from 2011 to 2014".

This might mean the same thing but it is very confusing.

I think Discussion points such as "Rather, they may be very busy with earning money and only have time to have a quick lunch or dinner, and therefore may be more likely to consume fast food." are very speculative and should be validated with other studies.

The study's main limitation, that it is based on observational data and therefore there may be unobserved confounders should be discussed.

Author Response

  1. Thank you for letting me review this manuscript. The manuscript is well written in easily understandable. The presented data is rather old and statistical analysis is unspectacular but acceptable.

Response:

Thanks. We added the limitation.

  1. I think it must be more clear what the study finds. The Discussion says "This study used more recent data of California adults to show that lower family income was positively associated with obesity from 2011 to 2014." In the conclusion, the authors write "The findings of this study show that family income was negatively associated with obesity among adults in California from 2011 to 2014". This might mean the same thing but it is very confusing.

Response:

Thanks for the suggestions. We revised to make the statement more consistently.

  1. I think Discussion points such as "Rather, they may be very busy with earning money and only have time to have a quick lunch or dinner, and therefore may be more likely to consume fast food." are very speculative and should be validated with other studies.

Response:

Thanks for the suggestions. We revised accordingly.

  1. The study's main limitation, that it is based on observational data and therefore there may be unobserved confounders should be discussed.

Response:

Thanks for the suggestions. We revised accordingly.

Round 2

Reviewer 2 Report

No command